# Optimizing the Ion Conductivity and Mechanical Stability of Polymer Electrolyte Membranes Designed for Use in Lithium Ion Batteries: Combining Imidazolium-Containing Poly(ionic liquids) and Poly(propylene carbonate)

**DOI:** 10.3390/ijms25031595

**Published:** 2024-01-27

**Authors:** Nataliya Kiriy, Sezer Özenler, Pauline Voigt, Oliver Kobsch, Jochen Meier-Haack, Kerstin Arnhold, Andreas Janke, Upenyu L. Muza, Martin Geisler, Albena Lederer, Doris Pospiech, Anton Kiriy, Brigitte Voit

**Affiliations:** 1Leibniz-Institut für Polymerforschung Dresden e.V., Hohe Str. 6, 01069 Dresden, Germany; 2Department Chemistry and Polymer Science, Stellenbosch University, Matieland 7600, South Africa; 3beeOLED GmbH, Niedersedlitzer Strasse 75c, 01257 Dresden, Germany; 4Organische Chemie der Polymere, Technische Universität Dresden, 01062 Dresden, Germany

**Keywords:** Li ion battery, solid polymeric electrolyte, imidazolium-containing polyacrylate, poly(propylene carbonate), plasticizer, ion conductivity, Li charge transference number

## Abstract

State-of-the-art Li batteries suffer from serious safety hazards caused by the reactivity of lithium and the flammable nature of liquid electrolytes. This work develops highly efficient solid-state electrolytes consisting of imidazolium-containing polyionic liquids (PILs) and lithium bis(trifluoromethane sulfonyl)imide (LiTFSI). By employing PIL/LiTFSI electrolyte membranes blended with poly(propylene carbonate) (PPC), we addressed the problem of combining ionic conductivity and mechanical properties in one material. It was found that PPC acts as a mechanically reinforcing component that does not reduce but even enhances the ionic conductivity. While pure PILs are liquids, the tricomponent PPC/PIL/LiTFSI blends are rubber-like materials with a Young’s modulus in the range of 100 MPa. The high mechanical strength of the material enables fabrication of mechanically robust free-standing membranes. The tricomponent PPC/PIL/LiTFSI membranes have an ionic conductivity of 10^−6^ S·cm^−1^ at room temperature, exhibiting conductivity that is two orders of magnitude greater than bicomponent PPC/LiTFSI membranes. At 60 °C, the conductivity of PPC/PIL/LiTFSI membranes increases to 10^−5^ S·cm^−1^ and further increases to 10^−3^ S·cm^−1^ in the presence of plasticizers. Cyclic voltammetry measurements reveal good electrochemical stability of the tricomponent PIL/PPC/LiTFSI membrane that potentially ranges from 0 to 4.5 V vs. Li/Li+. The mechanically reinforced membranes developed in this work are promising electrolytes for potential applications in solid-state batteries.

## 1. Introduction

Electronic devices, such as cell phones, laptops cameras, etc., have become an integral part of the daily life of modern society. These devices need lightweight rechargeable batteries [1], providing strong working capacity and autonomous operation. Li ion batteries (LIBs) are the most widely used types of rechargeable batteries. They contain a negative and a positive electrode, e.g., Li/C and LiCoO_2_ [2], in liquid electrolytes, i.e., solutions of lithium salts in organic solvents [3,4,5], separated by a permeable membrane. Unfortunately, state-of-the-art Li batteries create serious safety hazards [6,7] caused by the flammable solvents and the high reactivity of the lithium salts used (mainly LiPF_6_) toward environmental oxygen and moisture, which causes thermal runaway and fires. One of the most frequent causes of accidents is a leakage of flammable electrolytes occurring upon unintentional damage, which results in fires and even explosions [8]. In recent years, a high number of LIBs have been recalled due to those hazards [9].

To circumvent safety issues, many research groups [10,11] focus on the development of solid electrolytes for Li batteries [12,13,14]. To be able to replace liquid electrolytes in Li batteries, solid electrolytes should have a high ionic conductivity, a wide electrochemical stability window, low flammability, excellent processability, sufficient flexibility, and high thermal stability. Moreover, they should not cause leakage after battery damage [15,16]. Furthermore, when solid electrolytes possess appropriate mechanical properties and suitable interfacial properties, they can additionally act as a separator between the electrodes in Li batteries [17,18] and replace classical membranes.

Solid polymeric electrolytes (SPEs) [19,20,21] are the most actively developed class of materials due to their potentially good processability and mechanical properties. One of the most important challenge is the creation of SPEs in which appropriately good mechanical properties and high ionic conductivity are successfully combined [22,23]. In Li batteries, the ions are able to move freely through the electrolyte solution, resulting in high ionic conductivity [24]. In SPEs, in contrast, the lithium ions are coordinated to parts of the polymer chain, and their diffusivity is strongly coupled to the dynamics of the polymer backbone [25,26]. The ion movement is realized by ion hopping from segment to segment in intra- and intermolecular steps. Thus, a higher segment mobility, for instance above the glass transition temperature (*T_g_*) or above the melting point enhances the ionic conductivity, whereas the conductivity drops drastically below *T_g_* in the glassy state. Usually, the ionic conductivity in liquid electrolytes is at least two orders of magnitude higher than that in all-solid-state polymer electrolytes [27].

Electrolytes based on mixtures of polyethylene oxide (PEO) and Li salts have been extensively studied over the last 40 years [28,29,30,31,32]. PEO is among the most popular polymer components of SPEs due to its interesting characteristics, such as high capacity in salt complexation, high ionic conductivity in an amorphous state, good corrosion resistance, an acceptable commercial cost (for linear PEO), mechanical flexibility, and chemical stability [15,32] and references herein. However, PEO also possesses some disadvantages, which limit its technological application. One of the significant drawbacks of PEO is a low ionic conductivity at ambient temperature in the 10^−7^ to 10^−8^ S·cm^−1^ range caused by its propensity to crystallize [15]. However, at temperatures above the melting point, PEO-based PEs demonstrate appreciable conductivity due to the crystalline-amorphous phase transition [32]. To suppress the adverse effects of crystallization, many efforts have been directed toward achieving an amorphous state of PEO by employing more complex polymer architectures (e.g., comb-like copolymers, random copolymers, block-copolymers, cross-linked networks) using polymer blends and plasticizers or through incorporation of oxide nanoparticles [29,31]. An important step was achieved along these lines, and room temperature conductivities of up to 10^−4^ S·cm^−1^ were reported [29]. However, measures inhibiting the crystallinity of PEO, such as the addition of plasticizers, simultaneously affect the mechanical properties of SPEs. Nevertheless, several papers report PEO-based SPEs with a high ionic conductivity at room temperature (RT) in combination with good mechanical properties. For example, Watanabe et al. [33] described preparation of membranes composed of electrospun PEO-grafted-polyimide (PI-g-PEO) nanofibers and LiTFSI having a high conductivity of 10^−4^ S·cm^−1^ at room temperature and a high elastic modulus of 93 MPa [33].

In addition to PEOs, polymeric ionic liquids (PILs) are considered to be an attractive class of electrolytes [34,35,36,37,38]. Zheng et al. used UV-crosslinking in the synthesis of a lithium-containing PIL containing propylene carbonate [39]. Gerbaldi et al. prepared a single-ion block copoly(ionic liquid)s as an electrolyte using a reversible chain transfer polymerization technique [40]. However, these above mentioned PIL-based electrolytes show poor ionic conductivity and electrochemical stability during charge/discharge cycles. To improve ionic conductivity and electrochemical stability, PILs were mixed with IL-based electrolytes [41,42]. However, these electrolytes suffer from poor mechanical properties, owing to the high content of liquid components. Hierarchical PIL-based solid electrolytes were proposed by Kang et al. [43]; however, multi-step synthesis of the electrolytes is a drawback.

Many attempts have been made to mechanically reinforce polymer electrolytes [15], e.g., by cross-linking polymer chains producing ionic networks. While cross-linking efficiently enhances mechanical strength, it decreases the ion conductivity by about two orders of magnitude [27] due to the restricted mobility of polymer segments entrapped in the network. For example, our group recently reported imidazolium-containing poly(ionic liquid) (PIL) networks prepared by crosslinking photopolymerization. One advantage of the reported PILs is their high redox stability in the 4.3 V voltage window, which is crucial for applications in Li batteries [44]. It was shown that the crosslinks improved the mechanical stability of the PIL samples as verified by temperature-dependent rheology measurements. Furthermore, while the non-cross-linked PILs are viscous liquids, the cross-linked PILs are mechanically stable, which makes the preparation of free-standing membranes possible. However, the price for the improved mechanical properties is a reduction in ionic conductivity by two orders of magnitude from 10^−5^ S·cm^−1^ to 10^−6^–10^−7^ S·cm^−1^ for non-crosslinked and cross-linked PILs, respectively.

Lodge [45] reported an elegant approach for achieving very high mechanical strength (an elastic modulus approaching 1 GPa) for SPE without the sacrifice of its ionic conductivity. Their approach utilized a polymerization-induced phase separation (PIPS) phenomenon to produce a diblock copolymer in which one block is a rigid, high mechanical strength insulator and the other block is a low glass transition ion conductor. The deblock copolymer is used to independently tune the mechanical strength and conductivity of the SPE. In that report, the authors demonstrated that nanostructured block polymers with a disordered bicontinuous morphology are particularly attractive as highly conductive, thermally stable, and mechanically robust solid electrolyte membranes. The effect was achieved by the long-range, isotropic continuity of domains having a high modulus and high ion conductivity. While this approach worked and represents an excellent example of proof-of-concept, the use of poly(ethylene oxide) (PEO) as the ionically conductive component suffers from insufficient conductivity and redox stability. In addition, a complicated block copolymer synthesis might be an issue for industrial implementation of the method.

More recently, Zhang et al. [46] reported the preparation of safety-reinforced composites comprising poly(propylene carbonate) (PPC) and cellulose-based materials with an elastic modulus of 25 MPa at room temperature (RT). In that paper, in addition to the appropriate mechanical properties of PPC-based cellulose-reinforced composites, the authors claimed achieving high ionic conductivities of up to 3.0 × 10^−4^ S·cm^−1^ at 20 °C for the composites and for pure PPC. This work attracted our attention for the well-balanced mechanical and conductivity properties of PPC and its composites. An additional attraction of PPC is that it is the polymeric analog of alkyl carbonates, which are solvents used in commercial lithium batteries, and could open the possibility for complexation of Li ions. Starting from that report, the present study aimed at the development of a polymer material having sufficient high ionic conductivity and mechanical strength, which enables the preparation of free-standing membranes and integration of the latter into electrochemical devices. Rather than attempting to optimize the mechanical stability and ionic conductivity in a single-component material, we develop a composite consisting of two components, each of which is responsible for enabling one main property. Conceptually, this strategy is similar to the one pursued by Lodge et al. [45]. However, in their work, the two components were combined in a single block copolymer structure, whereas we utilized a mixture of two immiscible polymers, intending a spontaneous phase separation to form a bicontinuous phase-separated morphology.

In this work, three ionic acrylate-based polymers having alkyl-spacer imidazolium side groups (PILs) with alkyl lengths of C = 5, C = 6, and C = 10 (designated as PIL-5, PIL-6, and PIL-10, respectively) (Figure 1), were synthesized and characterized. PPC was chosen as a reinforcement component to impart appropriate mechanical properties to the composite. Lithium bis(trifluoromethane sulfonyl)imide (LiTFSI) was used as the lithium conductive salt.

## 2. Results and Discussion

### 2.1. Study of the Pure PILs and of Those with LiTFSI

The PILs studied in this work have quaternized ethyl imidazolium groups in their structure that are different lengths in distance from the acrylic backbone (Figure 1). Previous work [27] demonstrated that this design allows a fine-tuning of the glass transition temperature (*T_g_*) of the resulting PIL. In particular, a reduced *T_g_* enables sufficiently high segmental motion of the attached ionic groups at low temperature. This, in turn, contributes to enhanced mobility of Li cations, which move from one ionic position to another during the ion conductivity process. In the present work, we prepared a series of PILs with alkyl spacer lengths of C = 5, C = 6, and C = 10 designated as PIL-5, PIL-6, and PIL-10, respectively. The molar mass determination of these polymers is challenging due to interactions with the typical column materials used for size exclusion chromatography (SEC). For this purpose, we applied thermal field-flow fractionation (ThFFF), which is based on an open-channel separation and avoids such interactions [47]. An additional challenge was the detection and reliable molar mass calculation using the usually applied refractive index (dRI) and multiangle light scattering (MALS) detection. The difficulties arose from inaccurate d*n*/d*c* values because of the hygroscopic behavior of the PILs. To solve that problem, ThFFF coupled offline to time-of-flight mass spectrometry MALDI-TOF MS was applied, a new method that has been reported only once so far as a proof-of-principle on polystyrene (PS) standards [48]. The molar masses determined with ThFFF-MALDI-TOF MS are given in Table 1. The values obtained are typical for chain growth polymerizations, although the dispersities *Ð* appear a bit small.

The results of *T_g_* measurements by differential canning calorimetry (DSC) and ion conductivity performed by electrochemical impedance spectroscopy (EIS) for the pure PILs are given in Table 1 (see also Appendix A). The *T_g_* values of the pure polymers were determined from the 2nd heating cycle in DSC to avoid the influences of solvent residues. As expected, the *T_gs_* decreased when the alkyl spacer length increased (from −42 °C to −46 °C for PIL-6 and PIL-10, respectively). The PIL-5 with the shortest spacer had the highest *T_g_* in the series at −34 °C. The increase in the spacer length by a factor of two for PIL-10 compared to PIL-5 resulted in a reduction of 12 K, while increasing the spacer by only one methylene group led to a decrease in *T_g_* by 8 °C (from −34 °C to −42 °C for PIL-5 and PIL-6, respectively). A larger than expected reduction of *T_g_* for PIL-6 can be explained by the lower molar mass compared to the other PILs (Table 1) as well as by the well-known odd–even effect. The ionic conductivities at RT are summarized in Table 1. They range between 10^−5^ S·cm^−1^ and 2.9 × 10^−5^ S·cm^−1^ for dry PILs. PIL-10 with the largest alkyl spacer showed three times higher conductivity than PIL-5 with the shortest spacer. This result corroborates with a general trend established for ionic polymers according to which the ion conductivity depends inversely on *T_g_*. In our case, the PIL with the longest spacer and the lowest *T_g_* had the highest chain mobility at a given temperature above its *T_g_*; therefore, more efficient hopping of charges from one post to another occurs [49].

It should be noted that the ionic conductivities given in Table 1 refer to “pure” polymer samples, i.e., to those containing only “intrinsic” ions, namely, cations covalently attached to the polymer and TFSI counter anions, whereas all residual ions (e.g., chloride and lithium) were extensively washed after the ion-exchange synthetic step. To function as lithium-ion conductors, these ionic polymers are expected to require mixing with lithium-containing salts.

The ionic conductivity in the presence of lithium cations was studied in detail using PIL-6, the polymer with an intermediate conductivity, as an example. Table 2 presents ionic conductivities of PIL-6 blends with 10, 20 and 30 wt% LiTFSI at RT and 60 °C (see also Appendix A).

The addition of LiTFSI reduced the ion conductivity by an order of magnitude for membranes with 30 wt% LiTFSI compared to the pure PIL-6 membrane. However, the ionic conductivity was restored for samples heated up to 60 °C. This behavior can be explained in terms of a “solidification” effect as well as the weak solubility of LiTFSI in PIL-6, which suppresses the mobility of chains and decreased the conductivity. When heated up to 60 °C, the membranes soften, and the solubility is enhanced, which restores the conductivity.

### 2.2. Study of the Tricomponent Membrane PIL/PPC/LiTFSI

To prepare mechanically reinforced ionically conductive membranes, a tricomponent blend of PIL, PPC and LiTFSI solutions at different ratios in acetonitrile were drop casted on Teflon molds (Appendix A) and dried. An immediate outcome of these experiments was that the addition of PPC had a positive effect on the mechanical properties of the films. While pristine PILs were sticky materials, tricomponent blends of PILs with PPC and LiTFSI were solids at RT. Thus, free-standing membranes could be easily prepared by peeling off the drop-casted films from glass or Teflon substrates. These prepared membranes allowed certain manipulations and retained their integrity during conductivity measurements and mechanical tests. In contrast, bicomponent PIL/PPC membranes have much worse mechanical properties as will be discussed below. In the next step, the ionic conductivities of blend membranes were measured (Table 3). Reference membranes containing only PPC and LiTFSI were also tested.

As seen from Table 3, the highest conductivity was shown by the membranes with the polymer having the longest alkyl spacer. Interestingly, tricomponent membranes, i.e., those containing the PPC reinforcing component, exhibited even slightly higher conductivity compared to bicomponent PIL/LiTFSI membranes. Furthermore, an increase of the content of the reinforcing PPC component from a 1/1 to 1/3 PIL/PPC ratio had only a weak effect on conductivity, suggesting that the PPC additive did not cause an adverse effect on the conductivity (Appendix A). At the same time, the bicomponent PPC/LiTFSI membranes (i.e., membranes containing no ionic polymers) showed a two orders of magnitude lower conductivity than all membranes containing the PILs, suggesting that the ionic polymer is a necessary component for achieving ion-conductive membranes. This result is contradictory to the reported high conductivities of PPC blends with LiTFSI [46]. For a fair comparison with the literature data, PPC/LiTFSI membranes reinforced by cellulose additives were prepared. In our experiments, the cellulose-reinforced PPC samples also gave significantly lower conductivities compared to the values of up to 3 × 10^−4^ S·cm^−1^ at 20 °C reported by Zhang et al. [46]. Cyclic voltammetry measurements reveal a good electrochemical stability of the tricomponent PIL/PPC/LiTFSI membrane in a potential range from 0 to 4.5 V vs. Li/Li+, as follows from a low background current below 3 × 10^−5^ A (Appendix A).

### 2.3. Study of the PPC-Containing Membranes Treated with Acetonitrile

In a detailed study of PPC-based membranes, we found a strong dependence of their conductivity on the content of residual acetonitrile remaining in the samples due to incomplete drying. To quantitatively assess the influence of the residual solvent on the conductivity of membranes, a series of PPC/LiTFSI and PIL-10/PPC/LiTFSI films prepared by drop casting were extensively dried in vacuum at elevated temperatures. For controllable placing of acetonitrile into the polymer films, the dry samples were exposed to acetonitrile vapor in a closed chamber for various times to accomplish swelling. The content of absorbed acetonitrile was estimated by measuring the samples’ weights at different times of exposure to acetonitrile vapor. The acetonitrile content in these experiments varied from 2 to 30 wt%.

It was found that although the extensively dried PPC/LiTFSI films were almost insulators (1 × 10^−8^ S·cm^−1^, Table 3), the PPC/LiTFSI membranes containing ~10 wt% of acetonitrile exhibited two orders of magnitude higher conductivity of 2 × 10^−6^ S·cm^−1^ (Table 4, #1).

Furthermore, saturation of the PPC/LiTFSI with acetonitrile corresponding to approximately 30 wt% boosted the conductivity up to 3 × 10^−3^ S·cm^−1^. Taking these results into account, it was supposed that the high conductivity data reported by Zhang et al. [46] referred to samples containing residual acetonitrile. In contrast, the membranes containing PILs exhibited appreciable conductivities in the 10^−6^ S·cm^−1^ range even after exhaustive drying (Table 3). We attributed this result to the low viscosity of the ionic polymers, which plasticize the membranes even in the absence of acetonitrile. In contrast, the PPC/LiTFSI membranes were solids at RT, and they showed low conductivity in the absence of plasticizing agents, such as acetonitrile or PILs. Although conductivities of PIL/LiTFSI membranes were not as sensitive to the presence of acetonitrile as PCC-based membranes, the intentional addition of acetonitrile appeared also a viable method to boost their conductivities. For example, the addition of 8 wt% of acetonitrile to PPC/PIL-6/LiTFSI membranes increased their conductivity up to 3 × 10^−4^ S·cm^−1^, whereas they exhibited an ionic conductivity of 3 × 10^−3^ S·cm^−1^ at saturation (Table 4, #5 and #6).

The positive effect of solvent additives on the ionic conductivity of electrolytes is not surprising [50,51] given that polymer networks swollen in solvents (gel electrolytes) have a conductivity that ranges from 10^−4^ to 10^−3^ S·cm^−1^, whereas liquid electrolytes possess the conductivity in range of 10^−3^ S·cm^−1^ at RT [52,53]. However, in both these systems, the solvent is the dominating component, which manifests itself given the poor mechanical properties of these materials and that solvent leakage easily occurs. In contrast, the solvent is the minority component in the membranes prepared here. Thus, the high conductivities in the 10^−4^ S·cm^−1^ range obtained in the present work on solid membranes containing only 8–15 wt% of acetonitrile is very encouraging, taking into account that the small amounts of the solvent have no detrimental effect on the mechanical properties of the membranes (Appendix A). However, it should be mentioned that a further increase in acetonitrile, such as up to 30 wt% to achieve an ionic conductivity of 10^−3^ S·cm^−1^ conductivity, resulted in very sticky and difficult to handle membranes.

The use of acetonitrile as electrolyte additive requires its electrochemical stability at the operational potential range of lithium batteries. The Yamada group [54] investigated in detail the reduction of the reactivity of acetonitrile against Li. They found that metallic Li slowly reacts when placed in pure acetonitrile; however, this reaction is completely suppressed in a super-concentrated (i.e., 4.2 M) solution of LiTFSI in acetonitrile, corresponding to 6 wt% acetonitrile in the solution. Other research groups [52,53] also used acetonitrile as a plasticizer for polymer electrolytes to achieve enhanced conductivities. However, Nilsson et al. [55] critically evaluated the stability of concentrated electrolyte solutions containing acetonitrile and concluded that the stability of acetonitrile-containing electrolytes is overestimated, especially when “realistic” components of lithium batteries are used. Particularly, they noticed that highly concentrated electrolytes had much improved electrochemical stabilities. Their reductive decomposition below ca. 1.2 V vs. Li/Li+ and the oxidative corrosion of an aluminium collector at ca. 4.1 V vs. Li/Li+ are issues. As such, the use of acetonitrile itself in realistic Li batteries should likely be avoided. Nevertheless, we evaluated feasibility of the “plasticizing approach” utilizing acetonitrile as a prototypical plasticizer, having in mind that it can be replaced by more stable solvents, such as alkyl carbonates.

Li plating-stripping experiments were performed for the PIL-10/PPC/LiTFSI membranes containing 8 wt% of acetonitrile. As seen in Figure 2, a relatively small overpotential of ~0.2 V is required for the reduction of lithium cations. In general, the Li plating-stripping curves indicate a sufficiently high mobility of Li^+^ ions through the membrane to enable efficient operation of the lithium battery.

The Bruce and Vincent potentiostatic polarization method [56,57] was applied for determination of the lithium transference number (*t_+_*), which is an important performance characteristic of electrolytes in lithium batteries. The *t_+_* characterizes the contribution of lithium ions in the whole ionic flux, and achieving better performance of lithium batteries requires high *t_+_* values [58]. However, the presence of other mobile ions in the electrolyte, such as counterions with respect to lithium, decreases *t_+_*. One strategy to increase *t_+_* is to graft the anions onto the backbone of the polymer chain as the mobility of the tethered anion approaches zero [59,60]. At the same time, immobilized anions largely suppress the ionic conductivity of lithium cations. The present work uses an alternative approach, which assumes the use of polymers with covalent immobilization of cations counterbalanced by bulky and weakly coordinating TFSI “intrinsic” anions, whereas LiTFSI is physically mixed with the ionic polymer. In this case, the mobility of both “intrinsic” TFSI anions (i.e., those which are inherent part of PILs) and those TFSI anions that are added with the LiTFSI salt is suppressed by hydrophobic and steric interactions with the polymer. However, the mobility of TFSI anions is not fully suppressed as it is in the polyanion-based electrolytes, which optimizes the “productive” contribution of the ionic flux of smaller and weaker interacting lithium cations.

To perform the *t_+_* measurements, a small constant potential is applied to an electrolyte between non-blocking lithium electrodes, which leads to a decrease in the initial current value until a steady-state value is reached. If no redox reactions involving the anions occur at the electrodes, the anion current will vanish in the steady-state, and the total current will be caused by the cations, which are represented by lithium cations in our case. The experiments were performed with PIL-10 membranes aged for 5 days in PPC/PIL-10/LiTFSI at a 1/1/0.6 wt/wt/wt ratio and containing 8 wt% of acetonitrile, see Figure 3.

The obtained *t_+_* of 0.29 (see Table 5) is not high, but it is sufficient to enable efficient operation of the lithium battery, providing that this value is achieved for the electrolyte having a high conductivity. As discussed above, by choosing the polycation-based design of the electrolyte, we intentionally sacrifice the *t_+_* value (which otherwise can be maximized by using polyanions) to maximize the overall ionic flux and, particularly, the conductivity of lithium ions.

### 2.4. Study of the Topography, Viscosity, and Mechanical Properties of Membranes

The PIL polymers employed and PPC have very different chemical structures and polarity, so a macrophase separation of their blends is expected. In this work, we aimed to employ the phase separation phenomenon to produce membranes with demixed PIL and PPC phases, which are responsible for the ionic conductivity and mechanical reinforcement, respectively. We assumed that the most preferable scenario to maximize both properties is the formation of interpenetrated networks in which domains of the same kind are well interconnected. This type of arrangement should favor the ionic current, whereas the formation of isolated PIL domains inside a less conductive PPC matrix should hinder ionic transport. In addition, disconnection of the PPC domains should have an adverse effect on mechanical properties of the membrane. As an extreme, severe macrophase separation may lead to the formation of layered structures, such as bilayers, where each layer represents pure PPC and PIL components. This kind of arrangement should favor conductivity if the layers are arranged perpendicularly to electrodes. However, this is suboptimal for mechanical properties because of possible delamination processes. In the next step, the morphological and mechanical properties of the blend membranes were investigated using different methods.

Figure 4a–c compares macroscopic appearance of the bicomponent PIL-10/PPC (Figure 4a,b) and tricomponent PIL-10/PPC/LiTFSI (Figure 4c) membranes prepared by drop casting and peeled off from a Teflon substrate using tweezers. While the tricomponent membrane is a robust and homogeneous object, the bicomponent membrane is a “Janus-type” bilayer composed of a solid PPC foil and a honey-like PIL-10 film. The bicomponent membrane does not maintain its integrity. Figure 4d is a scanning electron microscopy (SEM) image of the cross-section of the bicomponent membrane prepared by cryogenic ultramicrotomy, showing that the topmost PIL-10 layer (indicated with red arrow) partially flows out from the bottom PPC layer. In contrast, no obvious macroscopic phase separation is observed on the SEM image of the cross-section of the tricomponent PIL-10/PPC/LiTFSI membrane, reflecting the homogeneous distribution of the three components on a multi-micrometer scale (Figure 4e). However, submicrometer structuring is seen in the tricomponent membranes, as follows from atomic force microscopy (AFM) data (Figure 5).

The complex viscosity of the mixed membranes compared to the pure PIL-10 polymer is given in Table 6. The PPC membrane has the largest viscosity in the series with a value of 182.2 kPa at room temperature (Appendix A), and the viscosity decreases to 35.6 kPa at 60 °C when PPC is present in the molten state. Pure PIL-10 has a very low viscosity, both at room temperature and 60 °C (1.1 and 0.7 kPa s, respectively), indicating its liquid state in the whole temperature range (Appendix A).

The bicomponent blend membrane PIL-10/PPC at room temperature (Appendix A) has only an order of magnitude higher viscosity than pure PIL-10 and even lower viscosity at 60 °C, reflecting a very weak reinforcement effect. A reason for this finding involves severe macrophase separation of the membrane into the layered structure of weakly interacting PIL-10 and PPC layers. In this case, the viscosity of the membrane is mostly represented by the “weaker” liquid-like PIL-10 component.

Interestingly, LiTFSI caused a significant increase in viscosity of the blend membrane, so that the tricomponent PIL-10/PPC/LiTFSI membrane (Appendix A) exhibited 30 times higher viscosity at RT than the bicomponent membrane PPC/PIL-10. It is assumed that LiTFSI acts as compatibilizer, decreasing the size of the polymers’ domains and allowing for the PPC component to maintain the integrity of the whole membrane and reinforce it.

To shed more light on the origin of the reinforcement effect, the morphology of the blend membranes was studied by AFM. Figure 4 shows AFM topography and phase images of the PIL-10/PPC bicomponent film prepared on a Si wafer. The topography image shown in Figure 4f reveals large sub-micrometer holes of ~8 nm in depth (Figure 4h). The phase image taken at the same spot (Figure 4g) reveals a significant material contrast of about 40° between the materials inside the holes and the background (Figure 4i).

As the brighter features in the phase image correspond to a harder material and because PPC is the hardest component among the materials constituting the blend, we conclude that the component of the topmost layer is PIL-10, whereas PPC is an underlying layer observed only through the holes. As such, the AFM results revealed a perpendicular segregation and stratification of the bicomponent blend according to the previously formulated suggestion. The morphology and phase images of the tricomponent PIL-10/PPC/LiTFSI membrane on a Si wafer are shown in Figure 5.

In the topography image presented in Figure 5a, there are two types of materials with a difference in height of 80 nm (Figure 5c). The holes were not found as in the case of the bicomponent film where a bilayer structures was formed by dewetting, with PIL-10 on top of a PPC film. The phase image (Figure 5b) demonstrates the formation of a network of the more hard (bright color) material and the presence of more soft material (dark color) in the tricomponent membrane with phase contrast from 15° to 50°. Indeed, LiTFSI acts here as compatibilizer allowing the PPC component to maintain the integrity of the whole membrane.

The mechanical properties of the tricomponent PIL-10/PPC/LiTFSI membranes were tested using quantitative nanomechanical AFM (AFM QNM). Tapping mode AFM images performed at different applied forces of 25 nN and 0.5 nN are shown in Figure 6a,b, respectively. Investigating the same location in the sample using tapping with different forces allowed assessment of the elastic properties of the two surface components. The first scanning on the area of 10 µm × 10 µm was performed with a force of 25 nN. This resulted in a mean height difference value between the topmost and valley-like material of 12.5 nm. A second scan was performed at the same location but with a lower force, namely 0.5 nN, which revealed a mean height difference of 3.9 nm (Figure 6a,c and Figure 6b,d respectively). This behavior reflects the elastic nature of the membrane due to the presence of the liquid-like phase. AFM QNM testing enabled the measurements of Young’s modulus, which was determined to be 77 MPa and 116 MPa on average for materials in the topmost and valley regions, respectively (Appendix A). This level of mechanical strength fits well with the minimum requirements of for all-solid-state polymer electrolytes in LIB, as noted in a recent review [15], according to which a successful electrolyte must have a Young’s modulus greater than 30 MPa. The mechanical strength values obtained for PILs/PPC/LiTFSI membranes also compare well with those reported by others [33,45,46,61,62,63] (Table 7).

### 2.5. Study of the Thermal Behavior of Membranes Using DSC and TGA

The thermal behavior of the PIL-10, PPC, LiTFSI, and mixtures of these components was studied using DSC and TGA (Table 8; Appendix A). Each sample was measured in three consecutive DSC scans, and the comparison of which provided additional information about changes in the blend structure occurring during the previous scans. The glass transition temperatures, *T_g_*, of pure PPC gave values of 12, 13, and 25 °C, according to the first, second, and third heating runs, respectively (Appendix A). A slight increase in *T_g_* as measured in the second versus the first scan is likely due to evaporation of residual solvent occurring during the first heating to 100 °C. The mixture of PIL-10 and PPC shows two T*_g_* transitions at −40 °C and +30 °C, which are close to the corresponding *T_g_* of pure PIL-10 and PPC, respectively (Appendix A), confirming a severe phase separation in the bicomponent blend.

In the next step, bicomponent blends of the polymers with LiTFSI were investigated using DSC. The PPC with 30 wt% LiTFSI showed *T_g_* values of 7 °C and 16 °C as measured in the first and the second scan, respectively (Appendix A), which is comparable with the *T_g_* of pure PPC. In contrast, the third scan revealed a drastic decrease in *T_g_* to −50 °C, presumably reflecting the decomposition of PPC upon heating to 200 °C in the second cycle. The incorporation of LiTFSI into PILs does not considerably change their *T_g_* values (Appendix A which, as a representative example, compares DSC of PIL-10 with and without LiTFSI).

The tricomponent PPC/PIL-10/LiTFSI blend shows a single broad transition in the −30 to 20 °C range centered at −6 °C (Appendix A). The absence of transitions inherent to pure components, such as that observed in the bicomponent PIL-10/PPC blend, indicates a better miscibility of the polymers in the presence of LiTFSI. This observation corroborates with the assumption that LiTFSI acts as a compatibilizer of PILs and PPC in the tricomponent membranes. Indeed, as shown above, the bicomponent PPC/PIL-10 membranes possess poor mechanical stability and can be easily disintegrated from the matrix of pure components, whereas the addition of LiTFSI greatly improves the mechanical stability of the membranes. Furthermore, as shown in the rheological measurements, while the bicomponent PPC/PIL-10 membranes exhibit the viscosity of the pure PIL component, the viscosity increases significantly in the presence of LiTFSI. As noted in the case of the bicomponent PPC/LiTFSI blend, the third DSC scan of the tricomponent PIL-10/PPC/LiTFSI blend reveals a significant reduction in *T_g_* down to −45 °C, presumably due to the thermal decomposition of PPC during the second heating cycle of up to 200 °C. However, a milder heating of the tricomponent membrane up to 100 °C during the first cycle even somewhat increases the *T_g_* value of the blend (which could be due to removal of solvents residues), reflecting the absence of phase separation in this blend. This is a rather unexpected and very encouraging result, considering that the blend was heated significantly above the *T_g_* of the reinforcing PPC component.

Indeed, the bicomponent PIL-10/PPC membranes fabricated by drop casting on hydrophilic Si substrates are mechanically unstable, and the harder PPC middle layer could easily be detached from the topmost and underlying layers, both represented by the liquid PIL-10 component, using a needle (Appendix A).

As shown by TGA, decomposition of PPC occurs above 200 °C (Appendix A). Pure LiTFSI had a high thermal stability with no degradation signs until 380 °C (Appendix A). In the presence of LiTFSI, the thermal stability of PPC is reduced to 180 °C according to TGA (Appendix A, for PPC with 30 wt% LiTFSI). The degradation of the PIL-10/PPC (1/1 wt/wt) blend started at 225 °C, which is between the degradation temperatures of the pure components (Appendix A). The tricomponent PPC/PIL-10/LiTFSI (1/1/0.6) membrane started to degrade at 180 °C, similarly to that of the PPC/LiTFSI blend. However, generally, the thermal stability of the PIL-10/PPC/LiTFSI fulfils the basic requirements for components of lithium batteries [15].

## 3. Materials and Methods

### 3.1. Materials

Acryloyl chloride (96%, Alfa Aesar, Ward Hill, MA, USA), 5-bromo-1-pentanol (>95%, TCI, Tokyo, Japan), 6-bromo-1-hexanol (>95%, TCI), 10-bromo-1-decanol (>95%, TCI), 1-ethylimidazole (>98%, TCI), magnesium sulfate (MgSO_4_, >98%, Sigma Aldrich, St. Louis, MO, USA), silver nitrate (AgNO_3_, 0.1 M, Sigma Aldrich), triethylamine (TEA, 99%, Alfa Aesar, Haverhill, MA, USA), a,a′-azobis(isobutyronitrile) (AIBN), and poly(propylene carbonate) (PPC, Mw = 50,000 g/mol, Sigma Aldrich) were used as received. Lithium bis(trifluoromethane sulfonyl)imide (LiTFSI, 99%, IoLiTec Ionic Liquids Technologies GmbH, Heilbronn, Germany) was dried under vacuum at 110 °C for 24 h prior the use, separator Cellgard 2500 (Celgard LLC, Charlotte, NC, USA).

### 3.2. Synthesis of PILs

The synthesis of PILs was performed using a four-step procedure similar to the method reported by Yoshizawa et al. [64] (Appendix A).

#### 3.2.1. Synthesis of Compound **1**

The reaction was performed under argon atmosphere. Briefly, 1.2 equivalents of TEA in THF was added to 1 equivalent of the corresponding bromo-alcohol (5-bromo-1-pentanol, or 6-bromo-1-hexanol, or 10-bromo-1-decanol) and stirred at room temperature (RT) for 1 h. Afterwards, the reaction mixture was cooled to 0 °C, and 2 equivalents of acryloyl chloride were added. The mixture was stirred for 1 h at 0 °C and for 24 h at 50 °C. The resulting solid white precipitate of TEAxHCl that formed during the reaction was filtered. THF was removed in a rotary evaporator, and the solid residue was dissolved in water. The resulting Compound 1 with corresponding spacer lengths (x = 5 or x = 6, or x = 10) was extracted with diethyl ether and washed several times with deionized water. The organic phases were dried over MgSO_4_ and filtered. After evaporation of diethyl ether, the products were obtained as brown liquid (yield: 70–85%).

#### 3.2.2. Synthesis of Compound **2**

One equivalent of the respective Compound **1** was mixed with 2.0 equivalents of 1-ethylimidazole and stirred for 72 h at 50 °C. The received substances were purified by sequential washing with ethyl acetate and diethyl ether. The products (compounds **2**) were obtained as yellow, viscous, sticky liquids (yield: 60–85%).

#### 3.2.3. Anion Exchange of Halide Anions by TFSI Anions of Compound **2**

The halide to TFSI anion exchange reaction was performed with stirring of appropriate amounts of one equivalent of Compound **2** and two equivalents of LiTFSI in deionized water at 55 °C for 24 h. During the anion exchange reaction, phase separation occurred since the TFSI salts are insoluble in water. The products were washed with deionized water until lithium bromide could no longer be detected in filtrates using the AgNO_3_-test. After drying in a vacuum, Compound **2** with TFSI anion was obtained as yellowish to brownish oily substances with yields of 80–95%. The anion exchange reaction was confirmed by signal shifts in the 1H NMR spectra (e.g., proton signals of the imidazolium group in the spectra of the TFSI salts appear at lower chemical shifts compared to the bromide salts, Appendix A).

#### 3.2.4. Synthesis of PILs

Polymerization of Compound **2** with TFSI anion with corresponding spacers (x = 5 or x = 6, or x = 10) was carried out in an ethanol solution at concentration of 50 mmol/L in the presence of AIBN (2 mol%) relative to Compound **2** with TFSI anion. Oxygen was removed from the reaction mixture using 3 consecutive freeze/thaw cycles. The reaction mixture was then stirred for 24 h at 78 °C under Ar. The obtained PIL with a corresponding spacer (x = 5, x = 6, or x = 10) were purified from the monomer by rinsing with ethanol. The resulting products are isolated as brown viscous and sticky substances with yields of 50–80%. The structures of the obtained PILs were confirmed by ^1^H NMR in DMSO-d6 (Appendix A).

### 3.3. Preparation of the Samples for Electrochemical Measurements

#### 3.3.1. Pure PILs and Bicomponent PIL/LiTFSI Membranes

Pure PILs, as well as bicomponent PIL/LiTFSI membranes were drop casted inside Teflon rings and placed on the stainless steel electrode of a Swagelok cell (Appendix A). The bicomponent PIL/LiTFSI mixtures with 10, 20 and 30 wt% of LiTFSI were prepared by the mixing of respective monocomponent solutions for 6 h at 50 °C. After drop casting, the samples were extensively dried at 100 °C in a vacuum for 12 h.

#### 3.3.2. The PIL Tricomponent Membranes with PPC and LiTFSI

The solutions of PPC, PIL, and LiTFSI at a concentration 0.3 g/mL in acetonitrile with wt/wt/wt ratios of PPC/PILs/LiTFSI of 1/1/0.6, 1/1/0.4, 1/1/0.2, 3/1/1.2, 3/1/0.8, and 3/1/0.4 were prepared. Afterwards, 60–200 µL of the solutions (exact amount depends on the desirable film thickness) were drop casted on Teflon molds or soaked on Celgard 2500 separator, placed on a mold, and dried by gradually increasing the temperature from RT up to 100 °C for 24 h in a vacuum (Appendix A).

#### 3.3.3. Membrane Treatment with Acetonitrile

The procedure was carried out in a glove box. The pre-weighed membrane was placed on a Teflon mold in a Petri dish filled with acetonitrile and closed with a glass lid. The amount of acetonitrile absorbed by the membrane was determined after 30 min of exposure to acetonitrile vapor (Appendix A). To achieve a 2 wt% of acetonitrile, the membrane was stored for 6–10 min (Table 4, entry #4).

### 3.4. Electrochemical Measurements

All electrochemical measurements were performed using a Gamry potentiostat, Interface 1010. For different electrochemical methods, Swagelok cells (Swagelok Co., Solon, OH, USA) were utilized based on the following setups: symmetrical cell setup (Li0/PIL/Li0) for complex electrochemical impedance spectroscopy (EIS) and for potentiostatic polarization measurements (PPM); asymmetrical cell setup (steel/PIL/Li0) for plating-stripping experiments and for cyclic voltammetry (CV).

The potentiostatic impedance measurements were carried out using the following parameters: 1 MHz to 100 mHz at open circuit voltage with 25 mV AC current.

For pure PILs, which are liquids, Teflon rings with a height 0.1 cm and inner diameter 0.8 cm were used in EIS measurements to prevent short circuits (Appendix A). The thicknesses of the Teflon ring used was the same as the thickness of the membrane.

The ionic conductivity was calculated using Equation (1):Ϭ = d/RA(1)
where d is a sample thickness, R is the bulk resistance, and A is the cross-sectional area of the sample.

The bulk resistance of PILs and membranes were read from the high-frequency intercept of the Nyquist plot with the Z′ real axis. The equivalent circuits of the Nyquist plots of PILs and membranes are given in Appendix A and were built using Gamry Echem Analyst software 7.10.0. (Simplex method). The thicknesses of the PIL/LiTFSI membranes were determined using Equation (2):A = W/d/S(2)
where A is thickness, W is weight of the membrane in g, d is the density in g/cm^3^, and S is the area in cm^2^.

The experimentally determined value for the density of PILs is 1.1 g/cm^3^. The calculated areas are 0.785 cm^2^ and 0.502 cm^2^ for the samples without and with Teflon rings, respectively (Appendix A).

The thicknesses of free-standing PPC/PILs/LiTFSI membranes were measured using a digital caliper gauge from Carl Roth (stainless steel, range 0–150 mm).

The transference number (*t_+_*) for Li ions was determined using the Bruce-Vincent potentiostatic polarization method [65]. Potentiostatic polarization experiments were performed with an applied voltage of 10 mV and polarization time of 40,000 s. The transference number for the Li^+^ ions was calculated according to Equation (3):*t_+_* = *I_ss_* (ΔV − *I*_0_*R*_0_)/*I*_0_ (ΔV − *I_ss_R_ss_*)(3)
where *I_ss_* is the steady state current, *I*_0_ the initial current, ΔV is the applied potential, and *R_ss_* and *R*_0_ are the electrode resistances before and after polarization, respectively. *R_ss_* and *R*_0_ were determined by fitting model parameters with a suitable equivalent circuit (Appendix A) using Gamry Echem Analyst software 7.10.0. (Simplex method).

### 3.5. Methods

^1^H nuclear magnetic resonance (NMR) spectra were recorded on an Avance III 500 Spectrometer (Brucker Corp. Billerica, MA, USA) at ambient temperature. Dimethyl sulfoxide (DMSO-d_6_) was used as a solvent.

Thermogravimetric analyses (TGA) were carried out on a Q5000 (TA Instruments, Newcastle, DE, USA) under nitrogen at a heating rates of 10 K⋅min^−1^ in a temperature range of 30 to 800 °C.

Differential scanning calorimetry (DSC) measurements were performed on a DSC 2500 (TA Instruments, Newcastle, DE, USA) under nitrogen with heating and cooling rates of 10 K⋅min^−1^. PIL samples were measured by heating-cooling-heating cycles in the temperature range of −120 to 200 °C. The bi- and tricomponent membranes were measured in the three following regimes: 1st regime involved heating to 100 °C followed by cooling, 2nd heating to 200 °C followed by cooling, and 3rd heating to 100 °C.

Thermal field-flow fractionation (ThFFF) coupled offline with matrix-assisted laser desorption/ionization time-of-flight mass spectrometry (MALDI-TOF MS) was applied to determine the molar masses of the Im PIL using dimethylacetamide (DMAc) as the carrier solvent and for dissolution. After the ThFFF separation, fractions from the peak maxima were collected and analyzed using MALDI-TOF MS.

ThFFF experiments were carried out with a TF2000 system with the following channel dimensions: 45.6 cm tip-to-tip length, width 2 cm, 250 μm thickness, and spacer material made from Mylar A and Teonex by DuPont Teijin Films Ltd. The auxiliary instrumentation included an isocratic pump, degasser, auto-sampler, actively heated and cooled ThFFF channel, PN3621 21-angle light scattering (MALS) detector with a laser of the wavelength 532 nm, and PN3150 differential refractive index (dRI) detector (all by Postnova Analytics GmbH, Landsberg am Lech, Germany). The channel pressure was maintained between ≈7.0 and 10 Bar by installing a back-pressure tubing (inner diameter 0.001 mm) between the ThFFF channel and the MALS detector to avoid vaporization of the carrier solvent and to stabilize the dRI detector. The cold wall block of the channel was cooled using a liquid cooling circuit with a Unichiller 025-MPC (2.5 kW) refrigeration unit (Peter Huber Kältemaschinenbau GmbH (now AG), Offenburg, Germany). The samples were injected using 50 to 100 μL volumes for concentrations ranging from 5 to 10 mg mL^−1^. All data recording and analyses were performed using the TF2000 version of the NovaFFF software (Postnova Analytics GmbH, Germany).

Atomic force microscopy (AFM) measurements were performed on a Dimension Icon, AFM (Bruker, Billerica, MA, USA). The topography study and E-Modulus determination were done in Tapping Mode TESPA and Peak Force QNM mode. All measurements were conducted under ambient conditions.

Scanning electron microscopy (SEM) was performed using a NEON40 SEM (Carl Zeiss Microscopy Deutschland GmbH, Oberkochen, Germany). The cross-sections for scanning electron microscopy (SEM) were prepared by fracturing the membranes in liquid nitrogen and inspected at an acceleration voltage of 1 kV and 3 kV.

Rheological measurements were performed with a rotational rheometer ARES G2 (TA Instruments, USA). The chosen geometry for the thin film measurements was a parallel plate geometry with a 8 mm diameter and approx. 0.5–1 mm gap. All oscillating measurements were carried out in nitrogen gas atmosphere under active axial force matching in compression with a sensitivity of 0.05 N, a frequency of ω = 5 rad/s, and a shear strain of γ = 1%. The temperature range was between −100 and 100 °C with a heating rate of 10 K/min. The measurements were performed for the several samples, and the mean values were considered as the most reliable.

## 4. Conclusions

In this work, solid polymeric electrolytes (SPEs) were prepared from acrylate-based ionic liquid polymers (PILs) by incorporation of PPC and LiTFSI and tested for applications in lithium batteries. First, three PILs with alkyl spacer imidazolium side groups with a length varied from C5 to C10, were synthesized. The highest ion conductivity for neat PILs measured at RT of 2.9 × 10^−5^ S·cm^−1^ was achieved for PIL-10, having a decyl spacer; this is the minimum value required for the SPE. Afterwards, the ion conductivity of blends of PILs with LiTFSI was studied, revealing somewhat reduced conductivity at room temperature, most likely because of the restricted mobility of polymer chains and ions in the presence of LiTFSI. Nevertheless, at an elevated temperature of 60 °C, when the PILs/LiTFSI blends are in the molten state, the conductivity is restored to the level inherent for pure PILs. The incorporation of PPC has a positive effect on the mechanical properties of membranes. Although pristine PILs and their mixtures with LiTFSI are liquids, the PILs/PPC/LiTFSI films are solids. Thus, free-standing membranes can be easily prepared by pilling off the drop-casted films, and those films maintain integrity and allow various manipulations. TGA measurements confirm the thermal stability of the PILs/PPC/LiTFSI membrane up to 180 °C. Young’s modulus as determined using AFM is 77–100 MPa for the PIL-10/PPC/LiTFSI membrane, a value two orders of magnitude higher than required for a battery separator, which should exceed 30 MPa. In addition, the PIL/PPC/LiTFSI composition is compatible with the Celgard 2500 separator. This is a useful option for applications at elevated temperatures to increase membrane integrity and to prevent short circuiting, when PILs/PPC/LiTFSI is present in the molten state.

We found that an extensively dried PPC or its mixtures with LiTFSI are poorly conductive in the absence of PILs or solvents. The PPC component in the three component PILs/PPC/LiTFSI membranes only has a minimal adverse effect on the conductivity of the composite membranes, and extensively dried PILs/PPC/LiTFSI membranes show 10^−6^–10^−5^ S·cm^−1^ conductivity at RT. We attribute this result to the liquid nature of our ionic polymers, which plasticize the membranes even in the absence of any solvents. However, the ionic conductivity of membranes could be significantly optimized by the addition of a small amount of the acetonitrile as a solvent. Particularly, the presence of 8 wt% or even 30 wt% of acetonitrile in PILs/PPC/LiTFSI membranes boosts the ionic conductivity by 1–2 orders of magnitude to reach 10^−4^ S·cm^−1^ and 10^−3^ S·cm^−1^ at RT, respectively. Finally, CV measurements reveal the good electrochemical stability of PILs/PPC/LiTFSI membranes in both dry or acetonitrile-saturated states in the voltage range from 0 V to +4.5 V versus Li^o^. The Li^+^ charge transference number for the PILs/PPC/LiTFSI membrane determined by Bruce and Vincent potentiostatic polarization method was 0.29, which reflects a sufficiently high mobility of Li^+^ ions. However, under real conditions, the decomposition of acetonitrile is expected. Therefore, a more electrochemical stable plasticizer instead of acetonitrile has to be used to solve this issue. This work shows that the synthesized ionic polymers and mechanically reinforced membranes prepared on their basis are promising electrolytes for potential solid-state battery applications.

## Figures and Tables

**Figure 1 ijms-25-01595-f001:**
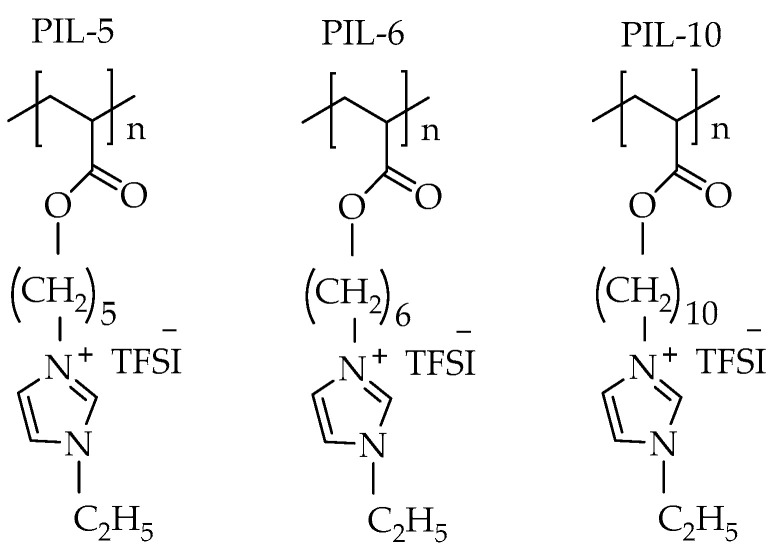
The structures of PILs studied in this work.

**Figure 2 ijms-25-01595-f002:**
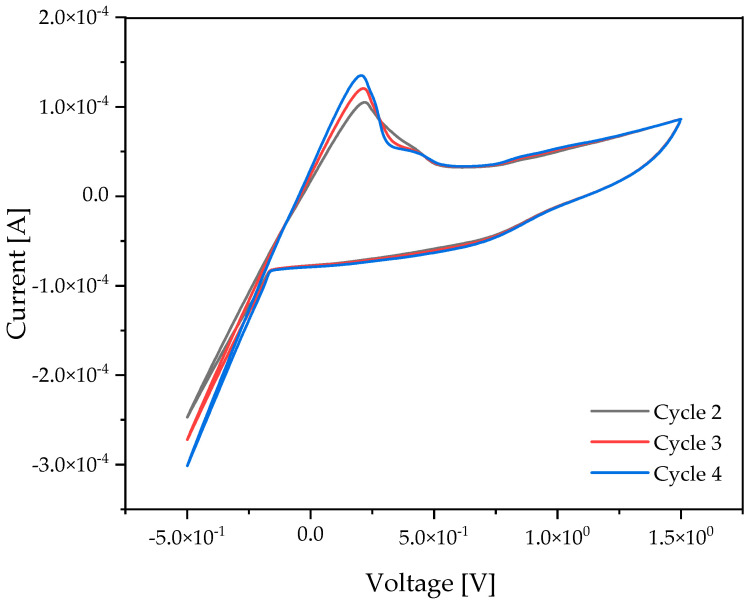
CV diagram of the PPC/PIL-10/LiTFSI at 1/1/0.6 wt/wt/wt and 8 wt% of acetonitrile vs. Li/Li+ in a Swagelok cell at 60 °C, scan rate of 1 mV/s.

**Figure 3 ijms-25-01595-f003:**
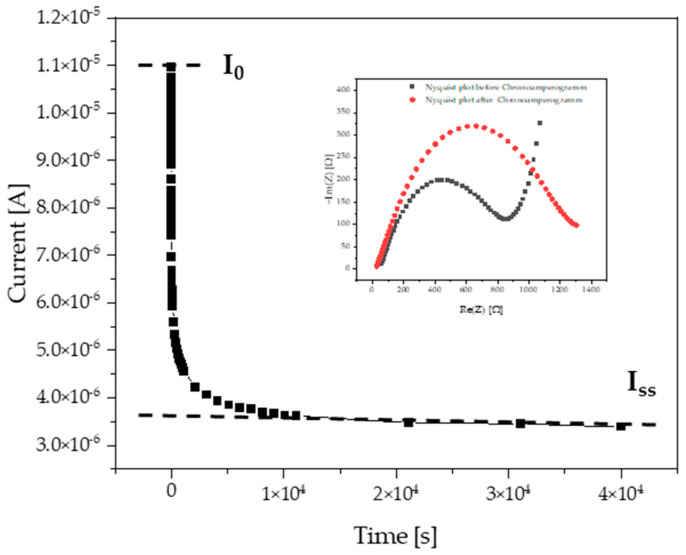
Chronoamperogram of the PPC/PIL-10/LiTFSI membrane at a 1/1/0.6 wt/wt/wt ratio recorded at an applied voltage of 10 mV (I_0_—the initial current, I_ss_—steady state current). The inset is the corresponding Nyquist plots before (black color) and after (red color) polarization chronoamperometry measurements. The experiments were performed at 60 °C.

**Figure 4 ijms-25-01595-f004:**
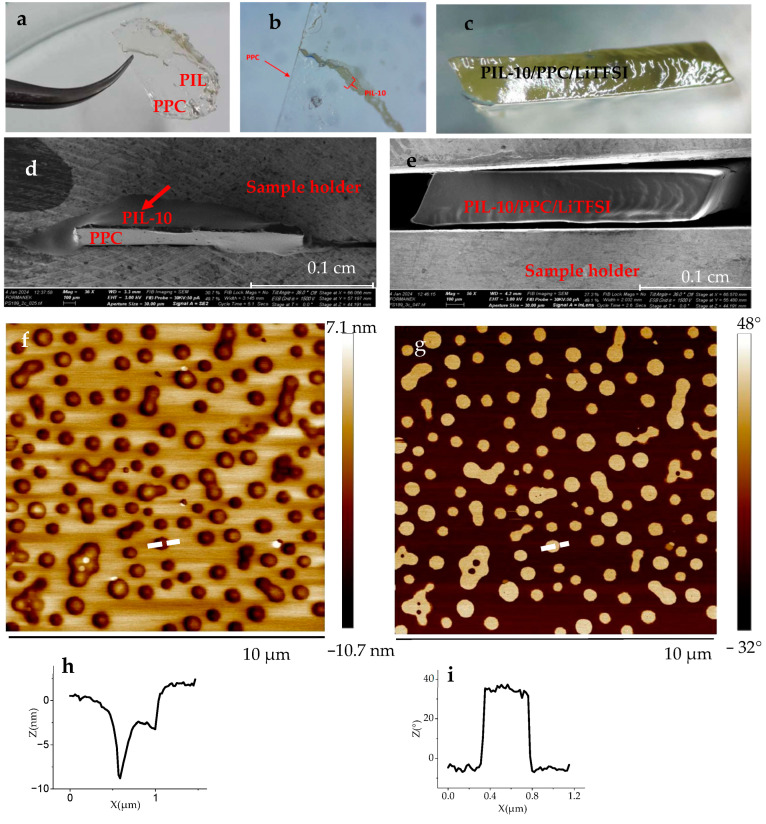
Photographs (**a**–**c**) of the PIL-10/PPC membrane (**a**,**b**) and the PIL-10/PPC/LiTFSI membrane (**c**). SEM images of the cross-sections of the PIL-10/PPC (**d**) and PIL-10/PPC/LiTFSI (**e**) membranes. AFM (**f**) height and (**g**) phase images with cross-section points (white dotted lines) and corresponding cross-sections (**h**) and (**i**) of PIL-10/PPC membrane (1/1) on the Si wafer.

**Figure 5 ijms-25-01595-f005:**
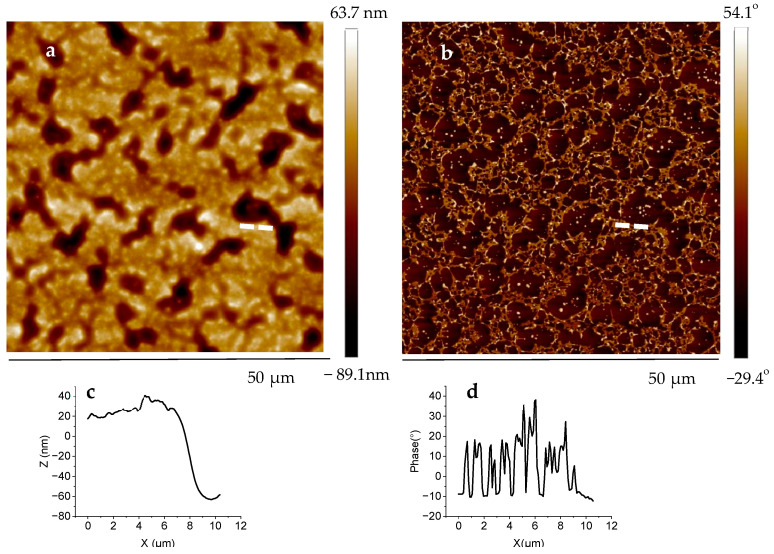
AFM (**a**) height and (**b**) phase images with cross-section points (white dotted lines), (**c**,**d**) cross-section of the PIL-10/PPC/LiTFSI (1/1/0.6) tricomponent membrane on a Si wafer.

**Figure 6 ijms-25-01595-f006:**
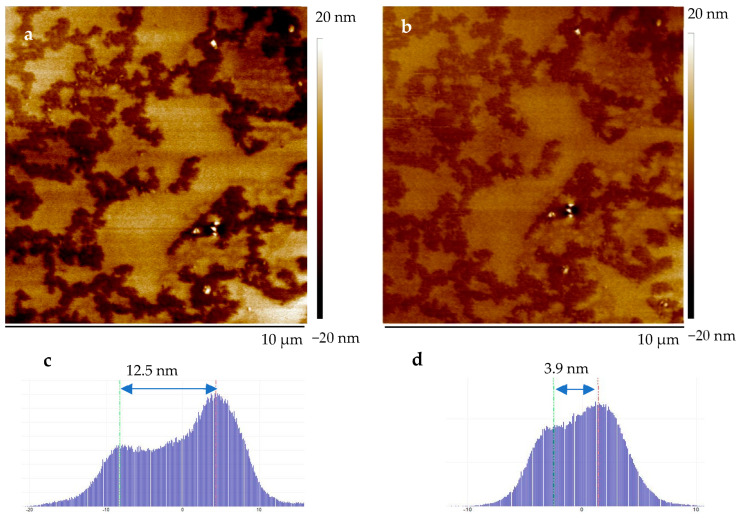
The AFM height image and height histogram of the tricomponent membrane PIL-10/PPC/LiTFSI (1/1/0.6 wt/wt/wt) prepared on Teflon mold: (**a**,**c**) with a tracking force of 25 nN; (**b**,**d**) with a tracking force of 0.5 nN.

**Table 1 ijms-25-01595-t001:** Average molar masses of PILs determined by ThFFF-MALDI-TOF MS, dispersities, *T_gs_*, and ionic conductivities of pure PIL measured at RT.

Sample	*M_n_* [kg·mol^−1^]	*M_w_* [kg·mol^−1^]	*Ð* (*M_n_*/*M_w_*)	*T_g_*, °C	Ionic Conductivity, S·cm^−1^
PIL-5	40.6 ± 2.4	53.7 ± 1.7	1.3 ± 0.04	−34	1.0 × 10^−5^
PIL-6	35.0 ± 1.1	43.8 ± 1.3	1.3 ± 0.01	−42	1.5 × 10^−5^
PIL-10	42.1 ± 0.8	53.8 ± 1.5	1.3 ± 0.06	−46	2.9 × 10^−5^

**Table 2 ijms-25-01595-t002:** Ionic conductivity of pure PIL-6 and PIL-6 with 10, 20, and 30 wt% of LiTFSI at RT and 60 °C.

Sample	Ionic Conductivity at RT [S·cm^−1^]	Ionic Conductivity at 60 °C [S·cm^−1^]
PIL-6	1.5 × 10^−5^	1.38 × 10^−4^
PIL-6 with 10 wt% LiTFSI	8.05 × 10^−6^	5.06 × 10^−5^
PIL-6 with 20 wt% LiTFSI	1.30 × 10^−6^	2.12 × 10^−5^
PIL-6 with 30 wt% LiTFSI	7.58 × 10^−7^	1.60 × 10^−5^

**Table 3 ijms-25-01595-t003:** Ionic conductivities of PIL/PPC/LiTFSI blends with the components in the ratio of 1/1/0.6 wt/wt/wt and PPC/LiTFSI (1/0.3) at RT (Appendix A).

Sample	Ionic Conductivity [S·cm^−1^]
PIL-5/PPC/LiTFSI	1 × 10^−6^
PIL-6/PPC/LiTFSI	2 × 10^−6^
PIL-10/PPC/LiTFSI	3 × 10^−6^
PPC/LiTFSI	1 × 10^−8^

**Table 4 ijms-25-01595-t004:** Ionic conductivities of PPC/LiTFSI (1/0.3 wt/wt), PPC/PIL-10/LiTFSI (1/1/0.6 wt/wt/wt) membrane, and PPC/PIL-6/LiTFSI (3/1/1.2 in wt/wt/wt) membranes at RT containing different amounts of acetonitrile (Appendix A).

Entry	Membrane	Amount of Acetonitrile [wt%]	Ionic Conductivity [S·cm^−1^]
#1	PPC/LiTFSI	10	2 × 10^−6^
#2	PPC/LiTFSI	20	1 × 10^−5^
#3	PPC/LiTFSI	30	1 × 10^−3^
#4	PPC/PIL-10/LiTFSI	2	1 × 10^−6^
#5	PPC/PIL-6/LiTFSI	8	3 × 10^−4^
#6	PPC/PIL-10/LiTFSI	30	3 × 10^−3^

**Table 5 ijms-25-01595-t005:** Electrode resistances R_0_ and R_ss_ before and after polarization, initial current I_0_, steady current I_ss_, and resulting *t_+_* for the PPC/PIL-10/LiTFSI membrane with a 1/1/0.6 wt/wt/wt ratio.

R_0_ [Ω]	R_ss_ [Ω]	I_0_ [A]	I_ss_ [A]	*t_+_*
890	1506	10.95 × 10^−6^	3.28 × 10^−6^	0.29

**Table 6 ijms-25-01595-t006:** Average viscosity values of the composite membrane and its separate components measured at 20 °C and 60 °C.

Sample	Viscosity at RT (20 °C), [kPa s]	Viscosity at 60 °C, [kPa s]
PIL-10/PPC/LiTFSI 1/1/0.6 wt/wt/wt/	242.8	3.5
PIL-10	1.1	0.7
PPC	182.2	35.6
PIL-10/PPC 1/1 wt/wt	8.3	0.6

**Table 7 ijms-25-01595-t007:** Comparison of the conductivity and mechanical properties of SPEs.

Sample	Young’s Modulus, [MPa]	Ionic Conductivity [S·cm^−1^] (at Temperature, [°C])	Reference
PPC/PIL-10/LiTFSI	~100	10^−6^ (RT); 10^−5^ (60)	This work
PI-g-PEO/LiTFSI	93	10^−4^ (RT)	[33]
PPC/cellulose/LiTFSI	25	10^−4^ (RT)	[46]
PEO/LiTFSI	0.45	10^−7^ (RT)	[33]
PIPS PS-b-PEO ^(a)^	~1000	10^−4^ (50)	[45]
PS-b-PEO	~50	10^−4^ (90)	[61]
PS-b-PEO-b-PS + IL	<1	10^−3^ (90)	[62]
PEO/LLZTO ^(b)^	4.73	10^−5^ (21); 10^−4^ (60)	[63]
PVDF-HFP/LLZTO ^(c)^	12.3	10^−7^ (21); 10^−6^ (60)	[63]

^(a)^ PIPS: polymerization-induced phase separation, PS: polystyrene. ^(b)^ LLZTO: ceramic electrolyte Li_6.4_La_3_Zr_1.4_Ta_0.6_O_12_. ^(c)^ PVDF-HFP: polyvinylidene fluoride-hexafluoropropylene.

**Table 8 ijms-25-01595-t008:** Thermal properties of blends and individual components as measured by DSC and TGA.

Sample	*T_g_* * [°C] (DSC)	T_start of dec.,_ [°C] (TGA)	T_maximum of dec.,_ [°C] (TGA)
PIL-10	−45	350	410
PPC	13	210	245
PIL-10/LiTFSI	−43	320	410
PPC/LiTFSI	16	180	206
PPC/PIL-10	−40/30	225	400
PPC/PIL-10/LiTFSI	−6	180	411

* The *T_g_* values were derived from the 2nd heating of the DSC curve.

## Data Availability

The data presented in this study are available upon request from the corresponding author.

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
