# Peer review of "Optimizing the Ion Conductivity and Mechanical Stability of Polymer Electrolyte Membranes Designed for Use in Lithium Ion Batteries: Combining Imidazolium-Containing Poly(ionic liquids) and Poly(propylene carbonate)"

_ijms, 2024, doi:10.3390/ijms25031595_

Round 1

Reviewer 1 Report

Comments and Suggestions for Authors

This paper describes the development of new SPEs based on IL copolymers of polyacrylates with varied spacer length. The study is well-planned and the charaxterization part addresses most of the material's properties. However, few comments can be addressed to improve the discussion or to reinforce the result outcomes:

1- The XRD analysis is not included. Although the Tg results shows the plasticizing effect, the crystallinity of the copolymers can provide more support to the the thermal properties. This is because, there should be a correlation with the amorphous phase with the mobility of the copolymers.

2- In the introduction part, it is advised to add few recent approaches that have been employed to alleviate the encountered problems in addition to use of SPEs,. e.g. Gel electrolytes. Pls. highlight the main advantaged of SPEs in more elaboration, too.

3- Authors did not compare the ionic conductivties of the tricomponent blends with the state-of-art ones. Pls. provide a brief discussion in the conclusion part to show how far/close the proposed PPC blend from the real application requirements in terms of conductivity. 

4- It is highly recommended to show the SEM of the cross-sectional fractured area to show the compatability between the polymer blends. The SEM images can be compared in the presence and the absence of LiTFSI to show the effect of the compatabilizer

Author Response

Response on the Reviewers’ and Editor’s requests, comments and suggestions

Dear Editor, we thank you for providing the possibility to revise our manuscript. In the revised manuscript, we addressed request, comments and suggestions from you and the reviewers. We provided a version of the revised manuscript for review where all changes are marked in red.

Below we place a point-to-point response to the reviewer comments:

Reviewer 1

This paper describes the development of new SPEs based on IL copolymers of polyacrylates with varied spacer length. The study is well-planned and the charaxterization part addresses most of the material's properties. However, few comments can be addressed to improve the discussion or to reinforce the result outcomes:

Response: We thank the reviewer for the generally positive evaluation of the manuscript!

1) The XRD analysis is not included. Although the Tg results shows the plasticizing effect, the crystallinity of the copolymers can provide more support to the thermal properties. This is because, there should be a correlation with the amorphous phase with the mobility of the copolymers.

Response: Thanks for the interesting suggestion. Indeed, XRD experiments may provide useful structural information of the samples, however, only in case of the presence of substantial crystallinity and/or order in the materials. However, the polymers used in the work are amorphous and the only crystalline component in the investigated compositions is LiTFSI. Although the degree of crystallinity of LiTFSI changes from the sample to sample, interpretation of these data would be questionable. Therefore, we did not perform XRD experiments.

2) In the introduction part, it is advised to add few recent approaches that have been employed to alleviate the encountered problems in addition to use of SPEs,. e.g. Gel electrolytes. Pls. highlight the main advantaged of SPEs in more elaboration, too.

Response: Thanks for the suggestion. In the revised manuscript, we extended the introductory part and added several new references (i.e., refs 28-43), addressing your suggestions.

3) Authors did not compare the ionic conductivties of the tricomponent blends with the state-of-art ones. Pls. provide a brief discussion in the conclusion part to show how far/close the proposed PPC blend from the real application requirements in terms of conductivity. 

Response: Thanks for a valuable suggestion. We discussed this point with regards to requirements to solid electrolytes formulated in the review article ref. 15 (lines 446-449 of the manuscript).

4) It is highly recommended to show the SEM of the cross-sectional fractured area to show the compatability between the polymer blends. The SEM images can be compared in the presence and the absence of LiTFSI to show the effect of the compatibilizer.

Response: Yes, we agree: SEM of the cross-sections could bring important information about the structure and internal morphology of the membranes. New comprehensive experiments along this line were performed and added to the manuscript (see new figure 4). These data unambiguously proved our hypothesis regarding the structure of the membranes.

In conclusion, we revised the manuscript according to the reviewers’ suggestions, which included performing of additional experiments and extensive editing of the text and hope that in the present form, it deserves publishing in the IJMS Journal. 

Best regards,

Anton and Nataliya Kiriy

Reviewer 2 Report

Comments and Suggestions for Authors

This work develops highly efficient solid-state electrolytes using imidazolium-containing polyionic liquids (PILs) and LiTFSI, addressing the challenge of balancing ionic conductivity and mechanical properties by incorporating poly(propylene carbonate) (PPC) as a reinforcing component. The resulting PPC/PILs/LiTFSI blends demonstrate good mechanical properties, high ionic conductivity, and potential for solid-state battery applications. It is interesting study and I would like to recommend it publication after the careful revisions as suggested below.

(1) Abstract should be self explanatory, every term should be spell out completely before using its abbreviation (e.g LiTFSI)

(2) Introduction section should be enhanced to highlight the novelty of the work. Latest literature reports should be cited. 

(3) Equivalent resistance circuit of Nyquist plot should be provided in the revised manuscript.   

(4) Conditions of experiments should be further explain, I could not get it how you performed the experiment. This is very important for the reproduction of work. 

(5) AFM figure placement is unadjusted , half figures are cut into page. You should revise it and provide professional placement of figures. 

(6) It is suggested to provide the basic characterization prepared samples via SEM, TEM. 

(7) TGA analysis of prepared samples should be provided for better understanding of thermal stability

(8) Tafel slope should be calculated for understanding the kinetics. 

(9) English of manuscript should be extensively improved. 

(10) References should be according to standard format of Journal. 

Comments on the Quality of English Language

English of manuscript should be extensively improved.

Author Response

Response on the Reviewers’ and Editor’s requests, comments and suggestions

Dear Editor, we thank you for providing the possibility to revise our manuscript. In the revised manuscript, we addressed request, comments and suggestions from you and the reviewers. We provided a version of the revised manuscript for review where all changes are marked in red.

Below we place a point-to-point response to the reviewer comments:

Reviewer 2

This work develops highly efficient solid-state electrolytes using imidazolium-containing polyionic liquids (PILs) and LiTFSI, addressing the challenge of balancing ionic conductivity and mechanical properties by incorporating poly(propylene carbonate) (PPC) as a reinforcing component. The resulting PPC/PILs/LiTFSI blends demonstrate good mechanical properties, high ionic conductivity, and potential for solid-state battery applications. It is interesting study and I would like to recommend it publication after the careful revisions as suggested below.

Response: We thank the reviewer for generally positive evaluation of the manuscript!

(1) Abstract should be self explanatory, every term should be spell out completely before using its abbreviation (e.g LiTFSI)

Response: Yes, we agree! It was corrected accordingly.

(2) Introduction section should be enhanced to highlight the novelty of the work. Latest literature reports should be cited. 

Response: Thanks for the suggestion, which coincides with the opinions of other reviewers. In the revised manuscript, we extended the introductory part, highlighted the novelty of the present work and added several new references, addressing your suggestions.

(3) Equivalent resistance circuit of Nyquist plot should be provided in the revised manuscript.   

Response:  In the revision Nyquist plots were added into the Supporting Information (new Figure S11 a-c).

(4) Conditions of experiments should be further explain, I could not get it how you performed the experiment. This is very important for the reproduction of work. 

Response:

(5) AFM figure placement is unadjusted , half figures are cut into page. You should revise it and provide professional placement of figures. 

Response: We apologize for the inconvenience of a partial file corruption that occurred upon the word file-to-pdf conversion. The problem is fixed in the revised manuscript.

(6) It is suggested to provide the basic characterization prepared samples via SEM, TEM. 

Response: We thank for the suggestion which share other reviewers. Because of that, SEM studies were performed (see new Figure 4 and discussions around).

(7) TGA analysis of prepared samples should be provided for better understanding of thermal stability

Response: Done: thermal analysis by TGA and DSC was performed for all materials and combinations thereof and these data and corresponding discussions are provided in the main text and

(8) Tafel slope should be calculated for understanding the kinetics.

Response: Kinetic studies are beyond the scope of this paper and likely will be a subject of upcoming paper

(9) English of manuscript should be extensively improved. 

Response: English was extensively revised throughout the manuscript

(10) References should be according to standard format of Journal. 

Response: Done:

In conclusion, we revised the manuscript according to the reviewers’ suggestions, which included performing of additional experiments and extensive editing of the text and hope that in the present form, it deserves publishing in the IJMS Journal. 

Best regards,

Anton and Nataliya Kiriy

Reviewer 3 Report

Comments and Suggestions for Authors

The authors have provided very important concepts on improving the ionic conductivity and mechanical strength of polymer electrolyte which are the two key factors that affect the electrochemical performance of solid-state battery using polymer electrolyte. Despite of the less popularity of PPC-based system, the quality and scope of the manuscript can cater to the taste of IJMS Journal, therefore, I would suggest it being published here after carefully revising my comments below:

1. The authors have to highlight the specific values (the ionic conductivity was increased from what value to what value, how the mechanical strength was increased, as well as the electrochemical window stability) in the abstract section. This would give readers an overview about the quality of the manuscript.

2. As the authors discussed the recent progress of SPE in the introduction part, however, the most popular SPE candidate-PEO-based electrolyte was not mentioned at all. Therefore, the authors are highly suggested to mention and compare the PPC with PEO based electrolyte in the introduction part to show the advantages and possibly disadvantages as well. Good examples for a PEO-based electrolyte reference is: energy storage materials, 57, 429-459, 2023; Nano Research Energy, 2023, 2: e9120050. https://doi.org/10.26599/NRE.2023.9120050; Ionics(2016)22:1259–1279

3. What are the rationals of choosing the PILs in Figure1? The authors need to include a separate and new paragraph to explain that.

4. It is sort of counter-intuitive that when the salt concentration was increased, the ionic conductivity of the SPE was reduced (Table 3, for example). How did that come? In principle, the ion dissolvation should be much higher when the salt concentration is reasonably higher. Therefore, the authors need to comment on this. 

5. The authors need to include a table to compare your results (ionic conductivity, mechanical strength etc.) with the state-of-art published results in the similar fields (using PPC as a polymer framework to make SPE).

6. In Figure 4a and 4b, why did the authors have a lot of dot spots instead of a relatively smooth regions? What happened? Because this is not the case for Figure 5a and 5b, as well as Figure 6a and 6b. The authors have provided very important concepts on improving the ionic conductivity and mechanical strength of polymer electrolyte which are the two key factors that affect the electrochemical performance of solid-state battery using polymer electrolyte. Despite of the less popularity of PPC-based system, the quality and scope of the manuscript can cater to the taste of IJMS Journal, therefore, I would suggest it being published here after carefully revising my comments below:

1. The authors have to highlight the specific values (the ionic conductivity was increased from what value to what value, how the mechanical strength was increased, as well as the electrochemical window stability) in the abstract section. This would give readers an overview about the quality of the manuscript.

2. As the authors discussed the recent progress of SPE in the introduction part, however, the most popular SPE candidate-PEO-based electrolyte was not mentioned at all. Therefore, the authors are highly suggested to mention and compare the PPC with PEO based electrolyte in the introduction part to show the advantages and possibly disadvantages as well. Good examples for a PEO-based electrolyte reference is: energy storage materials, 57, 429-459, 2023; Nano Research Energy, 2023, 2: e9120050. https://doi.org/10.26599/NRE.2023.9120050; Ionics(2016)22:1259–1279

3. What are the rationals of choosing the PILs in Figure1? The authors need to include a separate and new paragraph to explain that.

4. It is sort of counter-intuitive that when the salt concentration was increased, the ionic conductivity of the SPE was reduced (Table 3, for example). How did that come? In principle, the ion dissolvation should be much higher when the salt concentration is reasonably higher. Therefore, the authors need to comment on this. 

5. The authors need to include a table to compare your results (ionic conductivity, mechanical strength etc.) with the state-of-art published results in the similar fields (using PPC as a polymer framework to make SPE).

6. In Figure 4a and 4b, why did the authors have a lot of dot spots instead of a relatively smooth regions? What happened? Because this is not the case for Figure 5a and 5b, as well as Figure 6a and 6b. 

Comments on the Quality of English Language

English is very hard to understand. 

Author Response

Response on the Reviewers’ and Editor’s requests, comments and suggestions

Dear Editor, we thank you for providing the possibility to revise our manuscript. In the revised manuscript, we addressed request, comments and suggestions from you and the reviewers. We provided a version of the revised manuscript for review where all changes are marked in red.

Below we place a point-to-point response to the reviewer comments:

Reviewer 3

The authors have provided very important concepts on improving the ionic conductivity and mechanical strength of polymer electrolyte which are the two key factors that affect the electrochemical performance of solid-state battery using polymer electrolyte. Despite of the less popularity of PPC-based system, the quality and scope of the manuscript can cater to the taste of IJMS Journal, therefore, I would suggest it being published here after carefully revising my comments below:

Response: We thank the reviewer for the generally positive evaluation of the manuscript!

1. The authors have to highlight the specific values (the ionic conductivity was increased from what value to what value, how the mechanical strength was increased, as well as the electrochemical window stability) in the abstract section. This would give readers an overview about the quality of the manuscript.

Response: The abstract was edited accordingly

2. As the authors discussed the recent progress of SPE in the introduction part, however, the most popular SPE candidate-PEO-based electrolyte was not mentioned at all. Therefore, the authors are highly suggested to mention and compare the PPC with PEO based electrolyte in the introduction part to show the advantages and possibly disadvantages as well. Good examples for a PEO-based electrolyte reference is: energy storage materials, 57, 429-459, 2023; Nano Research Energy, 2023, 2: e9120050. https://doi.org/10.26599/NRE.2023.9120050; Ionics(2016)22:1259–1279

Response: Similar points were also raised by other reviewers. The introduction part was extended and additional references including the one mentioned by the reviewer were added.  

3. What are the rationals of choosing the PILs in Figure1? The authors need to include a separate and new paragraph to explain that.

Response: The motivation and strategy of the work were explained better

4. It is sort of counter-intuitive that when the salt concentration was increased, the ionic conductivity of the SPE was reduced (Table 3, for example). How did that come? In principle, the ion dissolvation should be much higher when the salt concentration is reasonably higher. Therefore, the authors need to comment on this.

Response: We thank for the very interesting question! The mentioned “intuitive” trend (i.e., direct proportionality) between the concentration of ionics and ionic conductivity is valid only for “simple”, “ideal” electrolytes, such as for liquid electrolytes at a low salt concentration regime.

The point is that the (ionic) conductivity is proportional not only to the number of charge carriers (ions) but also to their mobility and the contribution of the mobility on the conductivity is especially important for solid state polymeric electrolytes.

This effect is explained in the following way in the manuscript:

The addition of LiTFSI reduces the ion conductivity by an order of magnitude for membranes with 30% LiTFSI compared to pure PIL-6 membranes. However, the conductivity is restored for samples heated up to 60 °C. This behavior can be explained in terms of a “solidification” effect occurring upon addition of the salt, which suppresses mobility of chains and decreases conductivity. When heated up to 60 °C, the membranes soften again, which restores the conductivity.

The observed trend is explained in terms of a reduction of ion mobility upon increase of LiTFSI concentration. As it shown in the present work, LiTFSI reinforces PPC/PIL blends (the mentioned “solidification” effect), increasing their mechanical strength and shifting Tg to higher temperatures. Thus, increase of LiTFSI concentration reduces mobility of polymer chains, which in turn, decreases the ion mobility. Reducing  the Tg  by heating or/and by addition of acetonitrile increases (restores) mobility of polymer chains, which contributes to increased conductivity as a result of increased chain mobility.

5. The authors need to include a table to compare your results (ionic conductivity, mechanical strength etc.) with the state-of-art published results in the similar fields (using PPC as a polymer framework to make SPE).

Response: As this paper is not a review, we decided to compare conductivities only in the text and relatively briefly. In addition, it is difficult to address this request to full extent because of lack of mechanical characterization data in published works. Regarding conductivity, mostly quite high values approaching 10-4 S/cm are reported.

6. In Figure 4a and 4b, why did the authors have a lot of dot spots instead of a relatively smooth regions? What happened? Because this is not the case for Figure 5a and 5b, as well as Figure 6a and 6b. 

Response: As explained in our manuscript (pages 10, line 353 to page 11, line 364), “dots” on the topography AFM image Figure 4a – are holes in the PIL film through which the harder PPC material is “observed”. This kind of morphology is formed due to a phase separation of the bicomponent blend.

Morphologies of the tricomponent blends on Figures 5 and 6 are not smooth and they also represent phase separations. In general, a particular morphological appearance of a blend depends on many factors, such as nature of the components, their relative concentrations, as well as details of the film preparation conditions (solvent evaporation speed, temperature), underlying surface, etc. Because of this, even identical compositions may frequently form somewhat different film morphologies (having different appearance on AFM images), therefore different appearances for different blends are to be expected.

Rather than a particular shape of the morphological features (i.e., round-shaped holes or random-shaped islands), other morphological characteristics are more important for

description of the particular blend. Regarding our case, more important is a different degree of “phase contrast” (frequently called material contrast) on images Figures 4 vs 5: while in Fig 4, the contrast is sharp reflecting the phase separation to pure components (dark, soft PIL and light, harder PPC), a weaker contrast in figs 5 reflects a smaller difference in the hardness between the phase separated features suggesting a mutual “dissolution” of the three components due to the presence of LiTFSI in the blend.

The morphology of the membranes is discussed accordingly in the manuscript.

In conclusion, we revised the manuscript according to the reviewers’ suggestions, which included performing of additional experiments and extensive editing of the text and hope that in the present form, it deserves publishing in the IJMS Journal. 

Best regards,

Anton and Nataliya Kiriy

Round 2

Reviewer 1 Report

Comments and Suggestions for Authors

All comments have been addressed and the text was improved with new experimental results. The paper can be accepted in the current form.

Author Response

Many thanks for the recommendation.

Reviewer 2 Report

Comments and Suggestions for Authors

Thank you for incorporating my suggestions.

Author Response

Many thanks for the recommendation.

Reviewer 3 Report

Comments and Suggestions for Authors

Unfortunately, the authors did not carefully and properly resolve my comments. I will need to look at the further revised version again. 

For question #2, the authors did not fully address my comments. All the references need to be cited to make sure the introduction part is highly comprehensive, instead of solely and relatively randomly choosing one of them. 

In addition, the authors completely ignored by important comment #5. There are huge amount of work that can be found regarding the mechanical strength and ionic conductivity. However, the authors did not spend time working on it. Ionic conductivity etc. comparison table must be presented. Otherwise this work is not complete. You can use "N/A" for the parameters that you cannot find in the literature. 

Comments on the Quality of English Language

English needs to be improved. 

Author Response

Ref 3:Unfortunately, the authors did not carefully and properly resolve my comments. I will need to look at the further revised version again. 

For question #2, the authors did not fully address my comments. All the references need to be cited to make sure the introduction part is highly comprehensive, instead of solely and relatively randomly choosing one of them. 

Response:The above request does not look relevant - it is clearly impossible to cite all works related to that a hot topic, yet be comprehensive enough in the description of prior art, and being remained in a meaningful paper size. This is difficult to address even within a Review article format. Hope, the Editor  shares our opinion.

We did not choose "one of them" (references) - our paper contained 60 references!! most of which related to the introduction and they described different approaches and aspects of state of the art.

Nevertheless, we added the requested reference [Ionics 2016) and additionaly, other 4 references (marked in yellow). Also, we listed additionally advantages and disadvantages of PEO electrolites, as requested (marked in yellow).

Ref 3: In addition, the authors completely ignored by important comment #5. There are huge amount of work that can be found regarding the mechanical strength and ionic conductivity. However, the authors did not spend time working on it. Ionic conductivity etc. comparison table must be presented. Otherwise this work is not complete. You can use "N/A" for the parameters that you cannot find in the literature.

Response: we dissagree with a statement "huge amount of work" (regarding adressing mechanical properties). For example, recommended by the reviewer review article (ref 32 of Ramesh et al), the word "mechanical" (strength or properties) appears 38 times; nevertheless this review does not provide any!! value related to mechanical properties. Another recommended article [29], reviews 244 works but gives references only to a couple of paper which provided mechanical strength values (we cited those two refs in our manuscipt). 

Still, we added the requested table to show all papers which we were able to find which gave quantitative mech strength data (new Table 7).

Hope, we properly adressed requests of the Reviewer #3

Round 3

Reviewer 3 Report

Comments and Suggestions for Authors

Did not answer the questions at all. Have to change my opinion. Reject for a publication. 

Author Response

Dear Editor,

Honestly to say, we are in a big perplexity what exactly caused that reaction of the Reviewer 3...

We were fully sure that during the two revisions, we addressed all point raised by the reviewer 3 (as well as of the other two). At least, we did our best!  I emphasize, we were NOT at the position to ignore any the criticism of the reviewer 3. Even opposite: we fulfilled the suggestions despite we were disagreeing with those points.

Particularly the well-discussed Point #2 All the references need to be cited to make sure the introduction part is highly comprehensive, instead of solely and relatively randomly choosing one of them…„ or Point #5  “to include a table to compare your results with the state-of-art published results in the similar fields“ –

Indeed, such requirements would be appropriate for review articles, but not for original research papers. Nevertheless, we included a large paragraph in the introduction citing additional 20m refs, gave pros and contras of different approaches, and even added the requested comparison Table 8.

Taking this into account, could it be true the words:  “completely ignored my important comments” and “did not answer the questions at all”.

I think, there is no needs to repeat again our replies and if needed, please have a look on our response letters which accompanied the submission of the revised versions.

Another comment in addition: we would like to emphasize that the crucial requests of the Rev 3 are related to rather formal points –what should be in the introduction, in the abstract, comparison with literature and etc.; and there were no points against the scientific quality of the paper. Furthermore, in the very first report, the Rev 3 wrote:

“…The authors have provided very important concepts on improving the ionic conductivity and mechanical strength of polymer electrolyte which are the two key factors that affect the electrochemical performance of solid-state battery using polymer electrolyte. Despite of the less popularity of PPC-based system, the quality and scope of the manuscript can cater to the taste of IJMS Journal, therefore, I would suggest it being published …”

We ask the Editor to follow that suggestion related to scientific quality of the paper (which is in the same line with the opinions of other reviewers) and ignore critics regarding purely formal things, especially taking into account that we fulfilled them too.

Regarding the recommended revision of the abstract: the request of the Rev 3 was:

The authors have to highlight the specific values (the ionic conductivity was increased from what value to what value, how the mechanical strength was increased, as well as the electrochemical window stability) in the abstract section…

we added the achieved mechanic strength values already in the first revision; we stated

”While pure PILs are liquids, the tricomponent PPC/PIL/LiTFSI blends … having Young’s modulus in the range of 100 MPa. We did not show Young’s modulus of pure PILs because they are liquids.

 We added comparison of conductivities for the developed membrane versus PPC/LiTFSI in the current revision (from what value to what value), by saying

“The tricomponent PPC/PIL/LiTFSI membranes have an ionic conductivity of 10-6 S/cm at room temperature that is by two orders of magnitude higher conductivity of bicomponent PPC/LiTFSI membranes.”

We also include the information about the electrochemical window (new editions are marked in yellow). Also, the corresponding information was included in the text (lines 250-253) and added a new Figure S12 in SI.

But we do not provide in the Abstract any comparisons with literature data – if this was meant by the reviewer in his request from what value to what value - because this would make things too complicated…

Why? Example: ref 26 [Adv. Energy Mater. 2015, 5, 1501082] claims achieving conductivity of 10-4 S/cm for PPC/LiTFSI, which is 1000-10000 times higher than we were able to produce with the same system. We believe that the reported 10-4 S/cm is due to the presence of residual acetonitrile in their sample. Should we dispute/discuss this in the abstract? Or to put their data without explanation and  ruin our position? Instead, we leave them to answer on their results, but reserve the right to us not to cite any suspicious works. I think, this is a fair approach.

We hope on unbiased judging of our work and publishing it without further delays.

Sincerely yours,

Anton and Nataliya Kiriy